# Stereoselective Pharmacokinetics of Ketamine Administered at a Low Dose in Awake Dogs

**DOI:** 10.3390/ani14071012

**Published:** 2024-03-27

**Authors:** Gwenda Pargätzi, Alessandra Bergadano, Claudia Spadavecchia, Regula Theurillat, Wolfgang Thormann, Olivier L. Levionnois

**Affiliations:** 1Section of Anaesthesiology and Pain Therapy, Department of Clinical Veterinary Medicine, Vetsuisse Faculty, University of Bern, 3012 Bern, Switzerland; 2Clinic for Small Animals, Vetsuisse Faculty, University of Zurich, 8057 Zurich, Switzerland; 3Section of Occupational Safety, Health Protection and Environmental Safety, Department for Biomedical Research, University of Bern, 3008 Bern, Switzerland; 4Institute for Infectious Diseases, University of Bern, 3001 Bern, Switzerland; 5Department of Clinical Chemistry, Inselspital, Bern University Hospital, 3010 Bern, Switzerland

**Keywords:** dog, ketamine, norketamine, pharmacokinetic, stereoselective

## Abstract

**Simple Summary:**

Ketamine is commonly recommended as an adjunct to analgesic drugs, particularly in the postoperative care of hospitalized patients. Utilizing low doses is advised to target antinociceptive concentrations and obtain anti-hyperalgesia while minimizing undesirable effects. Infusion rates typically range from 0.002 to 0.2 mg/kg/min, largely based on human studies. However, there is evidence suggesting that plasma concentrations in dogs may be lower compared to humans under similar administration regimens. This study explores the stereoselective pharmacokinetics of ketamine administered alone at a low dose in dogs. The findings support the hypothesis that higher infusion rates may be necessary to achieve plasma concentrations comparable to those providing antinociception in humans, potentially leading to behavioural side effects. Significantly higher plasma concentrations were observed for S-norketamine compared to R-norketamine, while no difference was observed for the parent enantiomers.

**Abstract:**

The present study aimed to examine the stereoselective pharmacokinetics of racemic ketamine in dogs at low doses. The secondary aims were to identify associated behavioural effects and propose a ketamine infusion rate. The study was conducted on nine intact male beagles, with each dog undergoing two treatments (BOL and INF). For treatment BOL, an intravenous bolus of 1 mg/kg was administered over 2 min. The treatment INF involved an initial bolus of 0.5 mg/kg given over 1 min, followed by an infusion at 0.01 mg/kg/min for 1 h. Blood samples were collected for pharmacokinetic analysis. The median R/S enantiomer ratio of ketamine remained close to 1 throughout the study. Levels of S-norketamine were significantly higher than those of R-norketamine across all time points. Based on the collected data, the infusion rate predicted to achieve a steady-state racemic ketamine plasma concentration of 150 ng/mL was 0.028 mg/kg/min. Higher scores for behavioural effects were observed within the first five minutes following bolus administration. The most common behaviours observed were disorientation, head movements and staring eyes. Furthermore, employing ROC curve analysis, a racemic ketamine plasma concentration of 102 ng/mL was defined as the cut-off value, correlating with the occurrence of undesirable behavioural patterns.

## 1. Introduction

Ketamine is a well-established medication in both human and veterinary medicine for sedation and anaesthesia [1,2]. It has garnered attention for its antinociceptive and analgesic properties since the 1950s [3]. Sadove et al. reported ketamine’s analgesic activity even at subanaesthetic dosages, allowing a reduction in undesirable side effects due to dissociative states [4]. Later, the role of ketamine’s NMDA-antagonist activity was described [5], and the use of subanaesthetic intravenous infusion of ketamine as an adjuvant for pain therapy was more widely investigated [6]. In recent years, the use of subanaesthetic intravenous ketamine infusions has gained interest, especially in the context of multimodal pain therapy, with its potential to prevent wind-up, central sensitization and postoperative hyperalgesia in humans [7]. Veterinary studies have also recognized its benefits, contributing to the growing body of evidence [8,9].

In dogs, ketamine is often combined with other analgesic medications, using dosages typically extrapolated from human studies, spanning from 0.002 to 0.01 mg/kg/min [8,10,11]. The intravenous dosage recommended in the veterinary literature ranges between 0.002 and 0.017 mg/kg/min [12,13,14], which seems adequate to minimize undesired effects [15,16,17]. Human studies have reported analgesic plasma concentrations at approximately 150 ng/mL [18,19], while research on conscious dogs anticipated analgesic plasma concentrations to lie above 200 ng/mL, a concentration associated with the emergence of mild undesired effects [20,21]. Although an anti-hyperalgesic activity of ketamine has been hypothesized to occur at lower concentrations in dogs (100–150 ng/mL), this phenomenon remains inadequately characterized [9,20]. As long as more extensive studies are lacking to define the appropriate plasma concentration in dogs, it remains sensible to investigate if 150 ng/mL (based on data in humans) can be targeted.

The majority of pharmacokinetic investigations concerning ketamine in dogs have utilized significantly higher doses and, in many instances, have involved concomitant administration of other anaesthetic agents, consequently influencing its pharmacokinetic profile [21,22,23,24,25,26,27,28,29,30]. Nevertheless, when applying the reported kinetic variable to low-dose administration regimens in simulation analysis, it becomes evident that the estimated concentrations in dogs are consistently three to five times lower than those observed in humans [18,31]. This has been confirmed for a low infusion rate (0.01 mg/kg/min), as dogs reached notably lower plasma concentrations compared to humans [9]. These findings suggest that achieving the desired analgesic blood levels in dogs may necessitate higher dosages in comparison to human counterparts.

The present investigation provides further analyses from data collected in a previous larger study [9]. The first aim of the present investigation was to report the stereoselective pharmacokinetic profile of racemic ketamine administered at low doses in unmedicated dogs. The secondary aims were to identify behavioural side effects elicited by ketamine administration and to extrapolate a recommendation for ketamine infusion rate to target analgesic concentrations. The hypothesis was that a higher infusion rate than 0.01 mg/kg/min is necessary to target plasma concentrations above 150 ng/mL, but undesirable behavioural effects may be present, limiting the application of this dose regimen.

## 2. Materials and Methods

### 2.1. Animal Ethics

The in vivo experiments were approved by the local ethical committee for animal experimentation (Approval No. 2090), and all experiments were performed according to Swiss regulations and as described in the permission. Sample size analysis was performed based on the aim of investigating antinociception evoked by ketamine by means of differences in nociceptive withdrawal reflex thresholds [9].

### 2.2. Animals and Data Collection

The study was performed on nine (*n* = 9) intact male beagles, weighing between 7.6 and 12.7 kg (mean ± SD, 9.1 ± 1.5). The dogs were fasted for 6 h with permanent access to water before each experiment. After aseptic preparation of the skin, a 22 G intravenous (IV) catheter was placed in the right cephalic vein for ketamine administration; an 18 G IV catheter was connected to an extension set with a 3-way cap in the opposite cephalic vein for blood sampling. Each dog received two treatments (BOL and INF) with a minimum interval of 4 days. For treatment BOL, racemic ketamine (Narketan^®^10, Vetoquinol AG, Ittigen, Switzerland) was administered as a single IV bolus of 1 mg/kg over 2 min. For treatment INF, an initial bolus of 0.5 mg/kg was given over 1 min, followed by an IV infusion at 0.01 mg/kg/min for 1 h. For both treatments, T0 was defined as the end of the bolus administration. The drug was diluted with physiologic saline in a 50 mL syringe (concentration of 4 mg/mL) and strictly administered using a syringe pump to guarantee constant injection velocities. To ensure accurate delivery, a constant intravenous infusion of lactated ringer’s solution was administered at 2 mL/kg/h.

Venous blood samples were taken from the left cephalic vein in heparinized tubes shortly before ketamine administration and at 1, 2, 4, 8, 16 and 30 min after T0 in the treatment BOL or at 1, 20, 40, 60 and 80 min after T0 in the treatment INF. At each time point, the IV catheter was flushed with a heparinized solution and 2 mL of blood was removed; 3 mL of venous blood was then collected into clean heparinized tubes for testing (2 aliquots of 1.5 mL), and the first 2 mL of blood was re-administered to the dog. The total sampled blood volume was approximately 21 mL (2.5% of the estimated blood volume). All samples were immediately kept on ice and centrifuged soon after, and the plasma was stored at −80 °C until the determination of ketamine and norketamine enantiomers concentrations.

### 2.3. Plasma Level Analysis

Enantiomers of ketamine and of its active metabolite norketamine were measured in plasma using capillary electrophoresis [32]. Briefly, the assay is based upon liquid–liquid extraction of ketamine and norketamine from 1 mL of plasma followed by analysis of the reconstituted extract by capillary electrophoresis in the presence of a phosphate buffer (pH 2.5) containing 10 mg/mL highly sulphated β-cyclodextrin as a chiral selector [9]. For each enantiomer of ketamine and norketamine, the calibration range was between 0.05 and 2.5 μg/mL. Lamotrigine was used as an internal standard. Analyses were performed on a capillary electrophoresis analyser using a 50 μm ID uncoated fused-silica capillary of 45 cm effective length, an applied voltage of −20 kV and a cartridge temperature of 20 °C. The detection wavelength was 195 nm. The quantitation limit (limit of quantitation, LOQ) for all enantiomers was 2.5 ng/mL. Intraday precision data (n = 3) were <6.5 % and interday data (n = 4) were <9.2% for all enantiomers at concentrations ≥ 25 ng/mL. The time curves for R- and S-ketamine (R-ket, S-ket) and R- and S-norketamine (R-nor, S-nor) were obtained for each animal with both treatments. The enantiomer ratio (R/S) was calculated for each sampling time for visualization of stereoselectivity.

### 2.4. Pharmacokinetic Modelling

Each step of the pharmacokinetic modelling was performed with commercially available software (Monolix Suite^®^ v.2023R1, ©Lixoft, Antony, France). The following assumptions were made: the administration of racemic ketamine leads to an equal amount of administered R- and S-ket (this was not verified by measuring the isomers in the parent drug vial); there is no interconversion between R- and S-enantiomers [33]. Based on data fit, diagnostic plots, Akaike information criterion (AIC), the objective function value of the log-likelihood (OFV) and residuals analysis, the most suitable standard mammillary multi-compartmental model was determined for both R- and S-ket plasma concentrations of each individual (PKanalix^®^). This was performed to obtain initial estimates for the population modelling. Non-compartmental analysis for R-/S-ket concentrations was also performed to orient initial estimates (PKanalix^®^). Then, different combined parent-metabolite models of the population pharmacokinetic data were evaluated using the Stochastic Approximation Expectation–Maximization algorithm (SAEM, Monolix^®^). The final estimates were evaluated based on data fit, adequacy to initial estimates, diagnostic parameters, residual analysis, standard errors and visual predictive check graphics.

### 2.5. Prediction

A simulation was run to obtain a simple infusion regimen, allowing us to maintain a summative plasma level for ketamine enantiomers of 150 ng/mL (see justification in the discussion below) while minimising norketamine accumulation in plasma. The simulation was performed using both the formulas of a previously published algorithm [34] as well as the Simulx^®^ module of the Monolix software suite. Finally, a prediction of the context-sensitive half-time curve for ketamine based on the pharmacokinetic data set was performed.

### 2.6. Behavioural and Antinociceptive Data

The dogs were scored every minute for behavioural patterns during and after the administration of ketamine by an unblinded observer. The occurrence of seven predefined behavioural patterns was recorded using a checklist: #1 signs of disorientation, #2 repeated lateral head movements, #3 staring eyes, #4 irregular breathing pattern, #5 muscular tremors, #6 salivation or lip lickings and #7 excitement. At each evaluation, one point was attributed for each behaviour such that a score of 0 meant the absence of these abnormal behaviours, while a score of 7 meant the maximal occurrence of undesirable effects. In parallel, a non-invasive experimental test for the quantification of antinociception (nociceptive withdrawal reflex threshold evaluation) was performed in all dogs during the ketamine infusion (treatment INF). A more detailed description of the material, methods and results for these behavioural and antinociceptive assessments was addressed in a previous article [9] and will not be presented here.

### 2.7. Statistical Analysis

Data were presented as the median and interquartile range (IQR; 25–75th percentile). Analyses were performed with a dedicated statistical software (Sigmaplot 14.0, Systat software, Inpixon GmbH, Duesseldorf, Germany), and the significance was set at *p* < 0.05. The differences between R- and S-ket, as well as R- and S-nor, plasma concentrations were analysed by a 2-way ANOVA for repeated measures, followed by a post hoc Holm–Sidak test for pairwise comparison. The effect of time on enantiomer ratio and behavioural score was analysed with a Friedman ANOVA on ranks for repeated measures, followed by a post hoc Tukey test for pairwise comparison. Receiver operating characteristic (ROC) curve analysis was performed with the easyROC application [35] to evaluate which plasma concentration of racemic ketamine best allowed us to predict the reduced occurrence of undesirable side effects (cut-off for a behavioural score above 2, 1 and 0, respectively). Nonparametric analysis was performed to determine standard error and confidence interval [36], and the Youden index method was applied to determine the optimal cut-point [37].

## 3. Results

In one dog for treatment BOL and one for treatment INF, blood samples could not be obtained, so samples from eight dogs were analysed for each treatment.

### 3.1. Plasma Concentrations

During treatment BOL, the plasma concentrations of R- and S-ketamine reached a peak immediately after the end of injection, followed by a steep decrease over time (Figure 1). The concentration of S-ket was significantly higher than R-ket at 1 (*p* < 0.001) and 2 min (*p* = 0.001) after T0 with a difference of 16 (5–18) ng/mL (9% of the concentration).

The median R/S enantiomer ratio for ketamine was 0.95 (0.91–1.00) and did not differ significantly over time (Figure 1). The plasma concentrations of norketamine increased for 4 min after ketamine administration before decreasing. Levels of S-nor were significantly higher than R-nor at all time points, with a median R/S enantiomer ratio of 0.74 (0.64–0.86) (Figure 1). A similar picture was observed during treatment INF (Figure 2), while the median R/S enantiomer ratio for norketamine was 0.87 (0.78–0.90) (Figure 2).

### 3.2. Pharmacokinetic Analysis and Prediction

Both R- and S-ket were best modelled with a standard mammillary two-compartment model and R- and S-nor with a standard mammillary one-compartment model, issued from the central compartment of their respective parent drug (Figure 3).

The main steps to determine the best population models are described in Appendix A. The respective pharmacokinetic parameters are presented in Table 1. An example of prediction obtained from the population model is presented in Figure 4. Extrapolation from the pharmacokinetic model to target approximately 150 ng/mL of racemic ketamine in plasma suggests an initial bolus of 0.5 mg/kg over 1 min, followed by 0.033 mg/kg/min reduced after 1 h to 0.03 mg/kg/min (Figure 4, Appendix A). Infusion rate to steady state of racemic ketamine at 150 ng/mL was predicted at 0.028 mg/kg/min. At this infusion rate, this model predicted a context-sensitive halftime (50%) for racemic ketamine of 10 min that nearly did not change with a prolonged duration of infusion of 12 h.

### 3.3. Behavioural Scores

A significantly higher score for undesirable behavioural effects was observed within the first five minutes following the end of the bolus administration in both studies (Figure 5). The three most common behaviours observed were signs of disorientation, repeated lateral head movements and staring eyes. A hysteresis of 2 min was observed for all dogs between peak plasma concentration and peak behavioural score. Taking this time lag into account, ROC curve analysis (Figure 6) defined a plasma concentration for racemic ketamine of 138, 102 and 102 ng/mL as the best cut-off values correlating, respectively, with a behavioural score above 2 (AUC: 0.9; specificity: 87%; sensitivity: 82%), above 1 (AUC: 0.95; specificity: 89%; sensitivity: 97%) and above 0 (AUC: 0.96; specificity: 92%; sensitivity: 91%).

## 4. Discussion

This study demonstrates that the administration of the racemic mixture of ketamine at low doses in non-premedicated dogs did not result in stereoselective differences in the concentration time course between S- and R-ketamine. These findings are consistent with previous investigations of ketamine in dogs, both in vitro [38] and in combination with other medications [24,27,28,30]. It is worth noting that stereoselective pharmacokinetics can vary across different species [39] and have been observed for ketamine in horses [40].

Comparing the current findings with previously published data on ketamine pharmacokinetics in dogs is challenging due to the novelty of this study, which investigates ketamine in the absence of concomitant drugs. Furthermore, the individual examination of each enantiomer in this study complicates direct comparison with existing pharmacokinetic models for racemic ketamine. Notably, the observed plasma concentrations for both racemic ketamine and norketamine align closely with the anticipated values derived from previously published pharmacokinetic parameters [21,22,23,26,27,28,29,30]. These observations imply that the pharmacokinetics of ketamine may not be significantly impacted by the simultaneous administration of common anaesthetic drugs, as has been observed with fluconazole, for example, which, although reported, was not deemed clinically significant in the context of anaesthesia [29].

The final population pharmacokinetic models obtained for both S- and R-ketamine include a significant variability within subjects for several parameters across different occasions. Typically, interoccasion variability (IOV) accounts for fluctuations within an individual across different instances, while interindividual variability (IIV) reflects inherent distinctions among individuals. In Monolix^®^, these variabilities are treated independently, with individual parameter values on each occasion being mutually exclusive. This implies that IOV essentially functions as IIV. Nevertheless, in specific scenarios, such as when there are only a few instances per individual, it becomes inappropriate to estimate both IOV and IIV, leading to the retention of only IOV, as published earlier [41]. The implementation of IOV varies in other population pharmacokinetics software, making the estimation of IOV without IIV practically impossible. However, Monolix^®^ enables the simultaneous estimation of variability, capturing parameter differences both between individuals and within the same individual across different occasions [42]. In the present study, the two occasions referred to a single bolus and an infusion. It remains to be investigated if the variability observed is partly due to the mode of administration.

The context-sensitive half-time of 10 min reported in this study notably contrasts with the terminal half-times typically documented for ketamine. This is possibly attributed to differences in their respective definitions, as previously highlighted [43]. Given its significance in predicting the trajectory of effects induced by anaesthetic and analgesic agents, the low context-sensitive half-time underscores the potential for a rapid decline of the plasma concentrations following ketamine administration of a relatively short duration, as in the present study. Further exploration with prolonged infusion durations is warranted to verify the predicted sustained rapid reduction in plasma concentration over time.

Norketamine, recognized as an active metabolite of ketamine [44,45,46,47], demonstrates a discernible rise in plasma concentrations during ketamine infusion, with prolonged administration potentially leading to a slowdown in its elimination. However, the precise contribution of accumulated norketamine concentrations to the overall effect remains inadequately documented. Notably, the prolonged impact of norketamine has been suggested to potentially play a role in the enduring antidepressant effect of ketamine [47], although comprehensive investigation into this phenomenon is still warranted.

In human subjects, research indicates that plasma concentrations ranging between 100 and 150 ng/mL are necessary to effectively alleviate hyperalgesia and allodynia [15,19,20,48]. In the context of this study, the primary result is that the racemic ketamine plasma concentrations achieved from an intravenous infusion at 10 micrograms/kg/min in unmedicated dogs fall significantly below the anti-hyperalgesic plasma levels (150 ng/mL). Through the application of the obtained pharmacokinetic model, it is determined that an infusion rate of ketamine (racemic) at 28–33 micrograms/kg/min is necessary to attain this targeted level. For short procedures necessitating rapid attainment of the desired plasma concentration, an intravenous bolus of approximately 0.5 mg/kg is recommended. Nonetheless, the requirement for this specific plasma concentration in dogs, as well as the potential presence of analgesic effects at lower levels, remains insufficiently investigated. Notably, a separate study conducted on conscious dogs reported that analgesic plasma concentrations may be required to exceed 200 ng/mL [21], while the anti-hyperalgesic activity of ketamine has been suggested at concentrations close to 100 ng/mL [9].

Increasing the dose of ketamine to achieve optimal analgesic efficacy is often constrained by the potential emergence of undesirable behavioural side effects. While emergence delirium and dissociative states are recognized as undesirable outcomes of inappropriate ketamine anaesthesia in dogs [49], a comprehensive evaluation of the behavioural side effects associated with the administration of low-dose ketamine in canines has not been documented yet. In human subjects, adverse effects such as memory impairment, hallucinations, nightmares, dissociative states and confusion have been reported [50]. In the present study, no behavioural abnormalities were observed during the course of the infusion. However, in the initial minutes following bolus administration, transient signs of disorientation, repeated lateral head movements and a fixed gaze were noted in nearly all the subjects, with no lasting complications. Additional irregular breathing patterns, muscular tremors, salivation or lip licking and sporadic instances of excitement were observed. A correlation between the occurrence of abnormal behaviour and ketamine plasma concentration was noted, with the presence of at least three of these behavioural patterns being associated with concentrations exceeding 150 ng/mL. While further investigation is necessary, maintaining such potentially analgesic plasma concentrations appears challenging without the occurrence of at least a few mild behavioural side effects, such as transient disorientation, abnormal head movements, fixed gaze and lip licking.

Several limitations hinder the extrapolation of these findings to the canine population. The study involved a small cohort of only nine dogs with closely matched physical characteristics (male laboratory beagles). While it is customary in order to focus on a specific hypothesis, caution is warranted in extending the conclusions to all clinical scenarios. The predictive validity of the pharmacokinetic model is also compromised by the small sample size, and despite the application of a population model methodology, it cannot be considered a true representation of the entire canine species but solely for the type of subjects under investigation. Nevertheless, the visual predictive check and residual standard errors yielded satisfactory results. The selection of the blood sampling schedule was constrained to minimize both blood withdrawal and analysis costs. Although this approach does not offer detailed insights into precise metabolization dynamics and recirculatory effects, it proves adequate for predicting a clinical dose regimen. Another limitation lies in the assumption, without verification, of an equal amount of each enantiomer (S/R) in the parent drug vial. Lastly, incorporating female subjects in future studies would be beneficial to assess potential gender-specific differences in pharmacokinetics. To date, no comprehensive pharmacokinetic study in dogs has investigated gender disparities in ketamine enantiomers. Notably, in humans, male individuals exhibited higher ketamine plasma concentrations compared to females [51]. Similar observations were reported in mice [51], whereas an alternative study documented elevated concentrations in female rats [52]. Interestingly, the gender difference was nullified by orchidectomy, and subsequent testosterone replacement therapy reinstated the distinction [51]. The clinical relevance of such variations remains to be thoroughly investigated.

## 5. Conclusions

This study highlights the presence of stereoselective pharmacokinetics for norketamine but not for ketamine. Additionally, it underscores the requirement for a higher dose (threefold increase) of ketamine in dogs to attain plasma concentrations comparable to those deemed analgesic in humans. Importantly, it emphasizes the potential challenges associated with administering such doses without eliciting undesirable behavioural side effects. Consequently, there remains a clear need for targeted research aimed at understanding the dose-dependent antinociceptive properties of ketamine in dogs.

## Figures and Tables

**Figure 1 animals-14-01012-f001:**
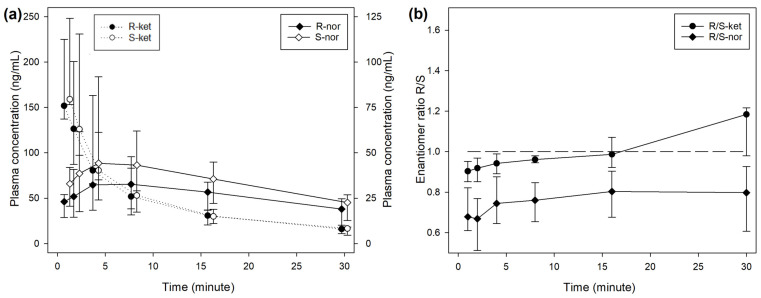
(**a**) Median (IQR) plasma concentrations of R- and S-enantiomers for ketamine (R-ket, S-ket) and norketamine (R-nor, S-nor) after intravenous administration of 1 mg/kg ketamine over 2 min (BOL treatment) in 8 beagles. Different scales are used for ketamine (left axis) and norketamine (right axis) to improve reading. Sampling time for R- and S-enantiomers was artificially slightly shifted to improve reading. (**b**) Median (IQR) R-/S-enantiomer ratio for ketamine (-ket) and norketamine (-nor).

**Figure 2 animals-14-01012-f002:**
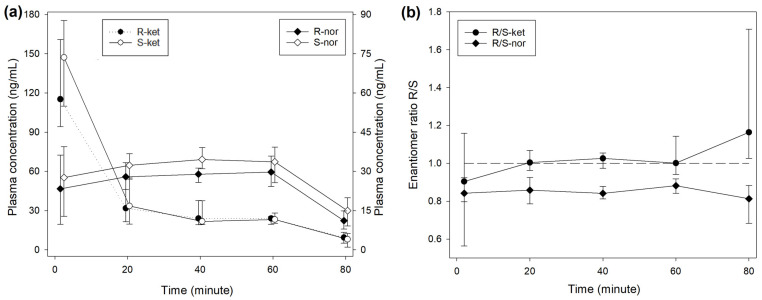
(**a**) Median (IQR) plasma concentrations of R- and S-enantiomers for ketamine and norketamine after intravenous administration of 0.5 mg/kg ketamine over 1 min, followed by 0.01 mg/kg/min for 60 min (INF treatment) in 8 beagles. Different scales are used for ketamine (left axis) and norketamine (right axis) to improve reading. Sampling time for R- and S-enantiomers was artificially slightly shifted to improve reading. (**b**) R-/S-enantiomer ratio for ketamine (-ket) and norketamine (-nor).

**Figure 3 animals-14-01012-f003:**
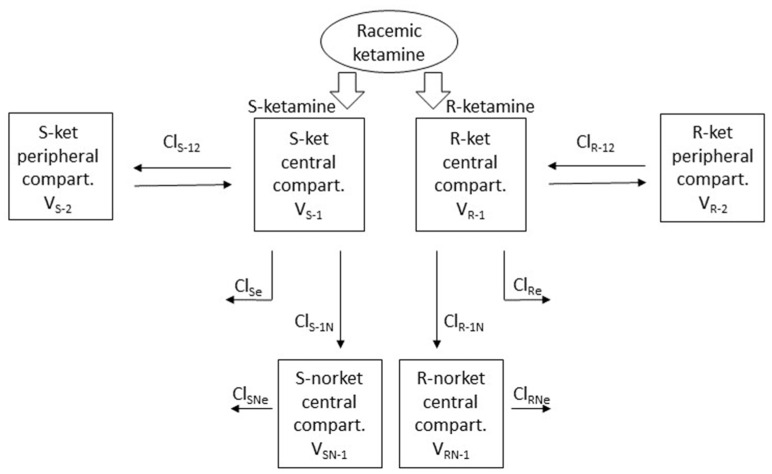
Schematic representation of the parent-metabolite mammillary multi-compartmental model with the best fit for the observed plasma concentrations of R- and S- ketamine and norketamine after administration of racemic ketamine alone at low dose in beagles.

**Figure 4 animals-14-01012-f004:**
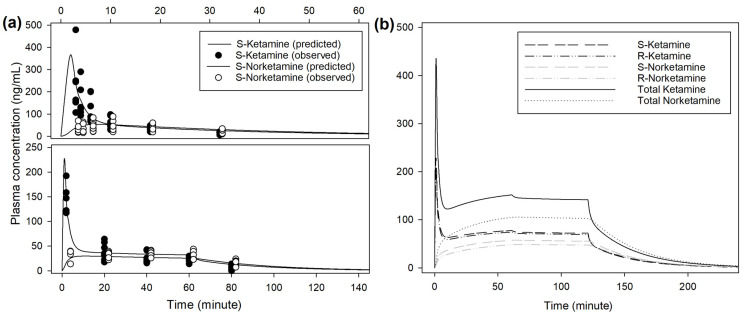
(**a**) Observed and predicted plasma concentrations of S-enantiomers for ketamine and norketamine after intravenous administration of 0.5 mg/kg ketamine over 1 min, followed by 0.01 mg/kg/min for 60 min (INF treatment, bottom) or 1 mg/kg ketamine over 2 min (BOL treatment, top) in 8 beagles. Sampling time for S-ket and -nor were artificially slightly shifted to improve reading. Scales for both plasma concentration and time are different for INF and BOL. (**b**) Predicted plasma concentrations of S- and R-enantiomers for ketamine and norketamine after intravenous administration of 0.5 mg/kg ketamine over 1 min, followed by 0.033 mg/kg/min for 60 min and 0.03 mg/kg/min for 60 min.

**Figure 5 animals-14-01012-f005:**
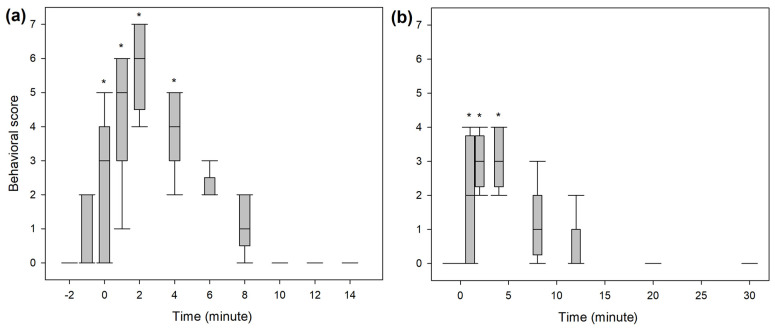
Whisker box plot (median, interquartile range, range) of the behavioural score assessed during intravenous administration of (**a**) 1 mg/kg ketamine over 2 min (BOL treatment) or (**b**) 0.5 mg/kg ketamine over 1 min, followed by 0.01 mg/kg/min for 60 min (INF treatment) in 8 beagles. * Statistically significant differences from baseline (*p* < 0.05, Friedman ANOVA on ranks for repeated measures followed by post hoc Tukey test for pairwise comparison).

**Figure 6 animals-14-01012-f006:**
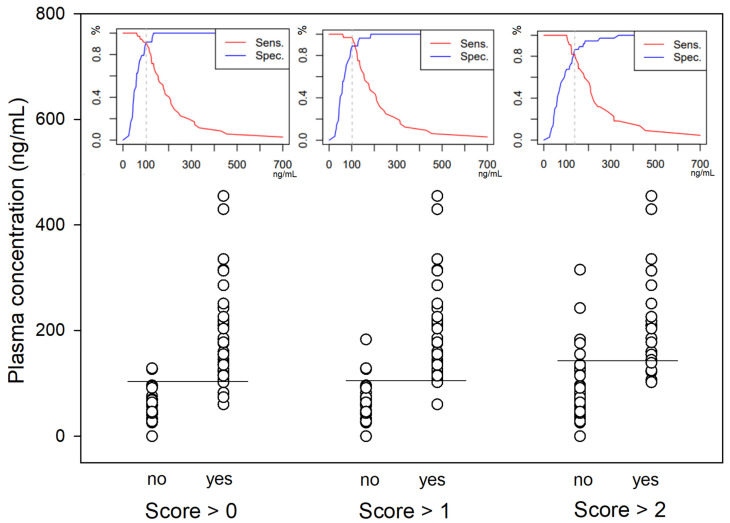
Scatter plot of the racemic ketamine plasma concentrations (n = 59) with a behavioural score above 0, 1 or 2. For each category, the best cut-off value is pictured by the horizontal line and the sensitivity/specificity diagram displayed above (racemic ketamine concentration in ng mL^−1^ on the x-axis, % on the y-axis).

**Table 1 animals-14-01012-t001:** Final estimates of pharmacokinetic parameters (see Figure 3) for the population parent-metabolite model of S- and R-ketamine in beagles receiving racemic ketamine alone at low intravenous doses.

	S-Ket			R-Ket		
	Fixed Effect	Covariate	Random Effect (γ IOV)	Fixed Effect	Covariate	Random Effect (γ IOV)
V_S/R-1_ (L kg^−1^)	1.05			1.52		
V_S/R-2_ (L kg^−1^)	2.82			4.14		
Cl_S/R-12_ (L kg^−1^ min^−1^)	0.18		0.88	0.16		0.92
Cl_S/R-e_ (L kg^−1^ min^−1^)	0.42	−0.13·BW	0.4	0.11		0.46
Cl_S/R-1N_ (L kg^−1^ min^−1^)	0.062		0.36	0.016	+0.11·BW	0.25
Cl_S/R-Ne_ (L kg^−1^ min^−1^)	0.079			0.091		

V, volume of compartment; Cl, clearance; IOV, intra-occasion variability; BW, body weight.

## Data Availability

The raw data supporting the conclusions of this article will be made available by the authors on request.

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
