# Peer review of "Stereoselective Pharmacokinetics of Ketamine Administered at a Low Dose in Awake Dogs"

_animals, 2024, doi:10.3390/ani14071012_

Round 1

Reviewer 1 Report

Comments and Suggestions for Authors

The study proposes the use of a ketamine pharmacokinetic model, developed with data collected from the present study, to recommend a ketamine infusion rate that would provide an estimated 150 ng/ml ketamine plasma concentration in dogs. This 150 ng/ml concentration is based on that used for human adjuvant pain management. I don’t understand the rational of this objective because similar ketamine plasma concentrations in humans and in dogs do not mean comparable effects between humans and dogs.

But, for that purpose, authors used two groups of dogs (the same nine dogs in each group, separated by at least 4 days interval): a single intravenous bolus of 1 mg/kg racemic ketamine administered over two minutes( blood samples at 1, 2, 4, 8, 16, 30 min after the bolus)  and an initial bolus of 0.5 mg/kg given over one minute, followed by an infusion at 0.01 mg/kg/min for one hour (blood samples at 1, 20, 40, 60 and 80 min after the bolus). The sampling is too scarce and does not allow the development of a thorough and precise bioavailability ketamine curve, which will obviously compromise the values of the intercompartmental model constants and, consequently, the validity of the pharmacokinetic model.

To obtain rigorous data for pharmacokinetic modeling, it is crucial to get the most detailed information possible from the bolus phase because it will provide the most important data regarding distribution and first metabolization of the drug. For example, the blood sampling could have been performed in short time intervals during the first two or three minutes after the bolus (for example, every 10 seconds in the first minute, and every 20 seconds in the next two minutes) followed by, for example, blood samples every 30 seconds until four minutes. Regardless of the ketamine bolus dose. And other blood collections should have been performed with increasingly time intervals during about 12 hours to obtain detailed information of the drug clearance.

The manuscript also lacks important information about the velocity of ketamine administration during the bolus, and how were the dogs distributed between groups and the equipment used for ketamine administration. This concern is present in figure 1a), where it can be observed a high IQR variation. Also due to the few blood samples used to model the bolus curve.

The use of just eight animals of the same breed of dogs is very restrictive in terms of extrapolation for the canine population in general. 

Author Response

Answers to the reviewers’ comments:

Thank you very much for your comments and advises, as well as for your positive evaluation of our work. We have adopted changes according to your comments. See details below. Changes are highlighted in yellow in the main text.

  • General comments:
    • Why to target 150ng/mL in dogs as in humans? (Reviewer #1)
      • As stated in the introduction, the plasma concentration targeted in humans may indeed not be appropriate in dogs. A few studies tend to support that dogs should have a similar effect response. However, if no other information is provided, it remains a better target than applying same dose despite PK differences. At least, knowing the PK behavior allows to adapt the dose to another target if this becomes to be defined in the future. In clinical setting, ketamine is regularly administered to dogs at 0.01 mg/kg/min with the idea to target approximately 150 ng/mL and we want to investigate if this is accurate. We have added a sentence in the introduction to clarify a bit more this point.
    • Sampling too scarce. (Reviewer #1)
      • Certainly, PK from 9 male beagles cannot be representative of the whole canine population. This is a strong limitation of the present study, and this is now better highlighted in the article.
    • Different and insufficient sampling times (Reviewer #1 / Reviewer #2)
      • Different sampling times have been applied for bolus and infusion regimens to better catch the elimination curve. Yet, more blood samples would of course improve the quality of the data. First, the study was initially not designed for an extensive PK study, therefore we agree that more blood samples could have been better planned. However, we believe that the most important parts of the kinetic time course have been caught (initial and secondary decay after the bolus, stabilization during infusion and comparison of the elimination curve). Second, the focus was on the aim to picture clinical dose regimen. Increasing blood sampling during the first minutes after the bolus as well as after the end of the infusion would provide essential information on early metabolization and recirculatory dynamics of ketamine but this was not of main interest here. There is little chance that the curve slopes between 2 and 6 minutes and 10 minutes after the bolus would change markedly if more samples would have been obtained. Therefore, sampling and analyses were performed more conservatively to limit blood withdrawal, equipment, and analysis cost. Yet, this is indeed a limitation of the study, and this has been added in the discussion.
    • Abstract (reviewer #2), lacks information about observed behavioral changes.
      • This has been added.
    • Abstract / Summary L. 33-37 (reviewer #3), infusion rates are presented in two different units of time - it may be clearer to readers to use either /min or /hour for both.
      • The unit has been changed everywhere in minutes for homogeneity.
    • Keywords (reviewer #2), sort alphabetically; words from the title are repeated.
      • Order has been adapted. We are aware that repetition of keywords from the title provides no added value, we could not find more appropriate proposals.
    • Introduction L. 82 (reviewer #2), I recommend adding a hypothesis.
      • This has been added.
    • 92 (Reviewer #1), how were the dogs distributed between groups?
      • There are no two groups. The same nine dogs were used at 2 occasions.
    • 98-105 (Reviewer #1), lacks important information about the velocity of ketamine administration during the bolus, and the equipment used for ketamine administration.
      • This has been added.
    • L 98 (Reviewer #2), how much blood (in mL) was taken?
      • This has been added.
    • L 124 (Reviewer #4), Please provide LOQ for the ketamine and norketamine isomers.
      • This was already included. It is highlighted now in the text.
    • L 134 (Reviewer #4), Did the investigators measure S(+) and R(-) ketamine isomers in the original drug vial to verify that the racemate had equal concentrations of both isomers?
      • No, this was assumed. This has been added in the method as well as in the discussion.
    • Figure 1, 2 (Reviewer 2), the points in the graphs mean the median, what do the whiskers mean?
      • These are IQR as indicated. It is more visible in the legend now.
    • Figure 3 (Reviewer #2), I guess the right cell should be "R-ket" instead of "S-ket"
      • That is right, thank you for noticing. This has been changed.
    • Figure 5 (Reviewer #2), I recommend highlighting significant differences in the graph
      • This has been added.
    • L 338 (Reviewer #3), I agree evaluating pharmacokinetics in female dogs may be advantageous, is there evidence from other species regarding sex specific pharmacokinetics of this drug?
      • A small discussion has been added with references.
    • L 346 (Reviewer #2), I recommend supplementing the limitation of the study
      • This section has been extended.

Reviewer 2 Report

Comments and Suggestions for Authors

The article is up-to-date, interesting and contributing. The use of ketamine for pain management in dogs is very important. I consider the new knowledge about the administration of ketamine in bolus or infusion to be beneficial.

Abstract

In the abstract, I lack information about observed behavioral changes, statistical methods used and clinical recommendations. I recommend to supplement.

Keywords

sort alphabetically; words from the title are repeated

Introduction

at the end of the Introduction, I recommend adding a hypothesis

Materials and Methods

L 98 – how much blood (in mL) was taken?

L 99-100 – why were not the blood samples taken at the same times?

Results

Figure 1, 2 – the points in the graphs mean the median, what do the whiskers mean?

Figure 3 – I guess the right cell should be "R-ket" instead of "S-ket"

Figure 5 – I recommend highlighting significant differences in the graph

Discussion

I recommend supplementing the limitation of the study

The manuscript is beneficial for the professional community as well as for practice. With regards of the above comments, I recommend accepting it after minor revisions.

Author Response

(The authors gave the same response as above.)

Reviewer 3 Report

Comments and Suggestions for Authors

Thank you for submitting this interesting research.

A few minor questions

Line 33 and line 37 - infusion rates are presented in two different units of time - it may be clearer to readers to use either /min or /hour for both.

Line 338 - I agree evaluating pharmacokinetics in female dogs may be advantageous, is there evidence from other species regarding sex specific pharmacokinetics of this drug?

Author Response

(The authors gave the same response as above.)

Reviewer 4 Report

Comments and Suggestions for Authors

Methods:

Please provide LOQ for the ketamine and norketamine isomers.

Results:

Did the investigators measure S(+) and R(-) ketamine isomers in the original drug vial to verify that the racemate had equal concentrations of both isomers?

Author Response

(The authors gave the same response as above.)

Round 2

Reviewer 1 Report

Comments and Suggestions for Authors

The authors have made an effort to address the limitations in the first version of the manuscript. However, the methodological limitations were note overcome which compromises the validity of the results.